# Plasma Metabolomics and Machine Learning-Driven Novel Diagnostic Signature for Non-Alcoholic Steatohepatitis

**DOI:** 10.3390/biomedicines10071669

**Published:** 2022-07-11

**Authors:** Moongi Ji, Yunju Jo, Seung Joon Choi, Seong Min Kim, Kyoung Kon Kim, Byung-Chul Oh, Dongryeol Ryu, Man-Jeong Paik, Dae Ho Lee

**Affiliations:** 1College of Pharmacy, Sunchon National University, Suncheon 57922, Korea; wlansrl@naver.com; 2Department of Molecular Cell Biology, Sungkyunkwan University (SKKU) School of Medicine, Suwon 16419, Korea; yj.eunice.jo@gmail.com; 3Department of Radiology, Gil Medical Center, Gachon University College of Medicine, Incheon 21565, Korea; sjchoi@gilhospital.com; 4Department of Surgery, Gil Medical Center, Gachon University College of Medicine, Incheon 21565, Korea; seongmin_kim@gilhospital.com; 5Department of Family Medicine, Gil Medical Center, Gachon University College of Medicine, Incheon 21565, Korea; zaduplum@gilhospital.com; 6Department of Physiology, Lee Gil Ya Cancer and Diabetes Institute, Gachon University College of Medicine, Incheon 21565, Korea; bcoh@gachon.ac.kr; 7Biomedical Institute for Convergence at SKKU (BICS), Sungkyunkwan University (SKKU), Suwon 16419, Korea; 8Institute of Quantum Biophysics (IQB), Sungkyunkwan University (SKKU), Suwon 16419, Korea; 9Department of Internal Medicine, Gil Medical Center, Gachon University College of Medicine, Incheon 21565, Korea

**Keywords:** nonalcoholic steatohepatitis, nonalcoholic fatty liver disease, metabolomics, machine learning, biomarkers

## Abstract

We performed targeted metabolomics with machine learning (ML)-based interpretation to identify metabolites that distinguish the progression of nonalcoholic fatty liver disease (NAFLD) in a cohort. Plasma metabolomics analysis was conducted in healthy control subjects (*n* = 25) and patients with NAFL (*n* = 42) and nonalcoholic steatohepatitis (NASH, *n* = 19) by gas chromatography-tandem mass spectrometry (MS/MS) and liquid chromatography-MS/MS as well as RNA sequencing (RNA-seq) analyses on liver tissues from patients with varying stages of NAFLD (*n* = 12). The resulting metabolomic data were subjected to routine statistical and ML-based analyses and multi-omics interpretation with RNA-seq data. We found 6 metabolites that were significantly altered in NAFLD among 79 detected metabolites. Random-forest and multinomial logistic regression analyses showed that eight metabolites (glutamic acid, cis-aconitic acid, aspartic acid, isocitric acid, α-ketoglutaric acid, oxaloacetic acid, myristoleic acid, and tyrosine) could distinguish the three groups. Then, the recursive partitioning and regression tree algorithm selected three metabolites (glutamic acid, isocitric acid, and aspartic acid) from these eight metabolites. With these three metabolites, we formulated an equation, the *MetaNASH* score that distinguished NASH with excellent performance. In addition, metabolic map construction and correlation assays integrating metabolomics data into the transcriptome datasets of the liver showed correlations between the concentration of plasma metabolites and the expression of enzymes governing metabolism and specific alterations of these correlations in NASH. Therefore, these findings will be useful for evaluation of altered metabolism in NASH and understanding of pathophysiologic implications from metabolite profiles in relation to NAFLD progression.

## 1. Introduction

Nonalcoholic fatty liver disease (NAFLD) is becoming a more burdensome disease with a global prevalence of between 20–40% depending on the population and up to 90% in obese people [1]. NAFLD is a multifactorial disease that includes it from mild steatosis to nonalcoholic fatty liver (NAFL) to nonalcoholic steatohepatitis (NASH), which may progress to cirrhosis and finally to hepatocellular carcinoma [2]. Up to 25% of patients with NAFL progress to NASH, with further progression to cirrhosis in 9–25% of patients with NASH [3,4]. Early diagnosis and therapeutic intervention are primary unmet needs in the management of NAFLD. Currently, a liver biopsy is required for the diagnosis of NASH and the monitoring of its progression, which is not widely applicable in routine clinical care for metabolic diseases and NAFLD [5]. Thus, methods for the noninvasive evaluation of NAFLD progression have been intensively studied. Imaging markers, blood biomarkers, and clinical scores have been introduced for these purposes, and some have been partially validated [6,7].

Using magnetic resonance (MR)-based techniques, we have constructed our own algorithm to predict the progression of NAFLD [6,8,9]. In our previous study [8] and an ongoing study thereafter, we found that the combination of MR imaging–proton density fat fraction (MRI-PDFF) and MR elastography-liver stiffness assessment (MRE-LSM) performed well in the identification of NASH with an area under the receiver operating characteristic (AUROC) curve value of 0.958 [95% confidence interval (CI) 0.909–1.000] at cutoff values of 16.8% and 3.8 kPa, respectively. However, MR-based techniques are expensive and not widely available to both patients and physicians. Current laboratory tests and clinical scoring systems have suboptimal accuracy in the identification of NASH [6].

Omics technologies and timely applications of machine learning (ML) and artificial intelligence techniques can rapidly process enormous amounts of data, providing an unbiased approach for the identification of biomarkers and therapeutic targets of NAFLD [10]. Thus, blood biomarker searching using high-throughput technologies, such as genomics, transcriptomics, proteomics, and metabolomics, has been an area of intensive research [11]. Metabolomics, which is the systematic study of small molecules in body fluids or tissues, holds promise for obtaining new insights into pathophysiologic processes, especially in metabolic diseases [12]. Therefore, to differentiate NASH from non-NASH, several reports have shown increases in some lipid metabolites including 11(S)-hydroxyeicosatetraenoic acid, which is a nonenzymatic oxidation product of arachidonic (20:4) acid [13], bile salt metabolites (glycocholate, taurocholate, and glycochenodeoxycholate), [14,15] chondroitin sulfate [16], amino acids (AAs) and their metabolites, [7,15] including γ-glutamyl peptides [15], branched-chain AAs (BCAAs) [7,15,17], glutamic acid, 2-hydroxyglutarate, the alanine/pyruvate ratio [7,15,17,18], and short-chain acylcarnitine metabolites including propionylcarnitine and 2-methylbutyrylcarnitine [15]. Decreased circulating levels of glutathione and related metabolites [15], serine and glycine [7,16], betaine [19], and lysophosphatidylcholines [7,15] in patients with NASH in comparison to non-NASH subjects were also reported. Additionally, a study reported different metabolome signatures according to subtypes of NASH [20]. However, these biomarkers have not been validated systematically and many biomarkers overlap between NAFLD and other metabolic diseases and between NAFLD stages [15,21]. Thus, in many metabolomics studies, NAFL and NASH could not be separated confidently [15]. More comprehensive and multiparametric approaches seem to be required to identify NASH and its progression accurately [11], considering that metabolic data are high-throughput and very complex and cover a wide range of analyte types and that the liver is a major regulator of whole body metabolism under various conditions. ML approaches in the interpretation of metabolomic data will be helpful for overcoming challenges in finding relevant metabolites for the accurate identification of NASH [22]. Here, we performed targeted metabolomics analyses with ML approaches for data interpretation to identify plasma biomarkers that differentiate NASH and non-NASH groups in a cohort that included healthy control subjects and patients with NAFLD at varying stages of progression. We also aimed to develop a diagnostic index for NASH based on metabolites that we found to be related to NAFLD progression.

## 2. Materials and Methods

### 2.1. Study Subjects

Metabolomic analyses were performed on a cohort of 86 subjects. The study subjects were eligible from a pooled cohort of four parent studies: (1) a study involving healthy control subjects without liver biopsy; (2) a study involving patients with NAFLD who volunteered for an MR-based NAFLD study, with some having liver biopsy data; (3) a study involving a bariatric surgery cohort with liver biopsy results; and (4) a study of living liver transplant donors with liver biopsy results. Through our previous study [8] and ongoing study thereafter, we established an algorithm combining MRI-PDFF and MRE-LSM parameters for the discrimination of NASH. Based on this algorithm, in the present study, the 86 study subjects were classified into healthy control (*n* = 25), NAFL (*n* = 42), and NASH (*n* = 19) groups. The healthy controls were required to have an MRI-PDFF less than 5% and normal results of liver function and other biochemical tests even without liver biopsy, including AST < 40 U/L and ALT < 35 U/L in males and AST < 40 U/L and ALT < 25 U/L in females. Study subjects were considered to have NASH if they had (1) an NAFLD activity score (NAS) ≥ 4 upon liver biopsy or (2) MRI-PDFF ≥16.1% and MRE-LSM ≥ 3.8 kPa in cases who did not receive liver biopsy. The other subjects with steatosis based on MRI-PDFF (≥5%) were considered to have NAFL. These subjects were determined eligible based on a study involving patients with NAFLD who volunteered for an MR-based NAFLD study, with some having liver biopsy data based on clinical indications. The ages of the study subjects were required to be between 19 and 70 years of age. Excessive alcohol consumption (>20 g/day for women and >30 g/day for men), evidence of another coexisting liver or biliary disease other than NAFLD, usage of drugs known for causing secondary hepatic steatosis within one year, and any conditions that could influence patient competence or participation as defined by the principal investigator were all exclusion criteria. The four parent studies were conducted in accordance with the Declaration of Helsinki and approved by the Institutional Review Board of the Gachon University Medical Center. All participants signed a written informed permission form, and all parent studies were logged on the website of the national center for medical information and knowledge (NCMIK) (https://cris.nih.go.kr (accessed on 16 June 2020)) in accordance with the International Clinical Trials Registry Platform.

### 2.2. Clinical and Laboratory Evaluation

Various clinical and laboratory data were collected in the studies, as detailed previously [8]. After an overnight fast, blood samples were collected on the same day or within days of the imaging studies or several days before liver biopsy to examine multiple markers and perform regular blood biochemical tests, which included liver function, a complete blood count with a platelet count, albumin, glucose, insulin, hemoglobin A1c (HbA1c), lipid panels, complement factors C3 and C4, and the enhanced liver fibrosis (ELF) test. Sample processing and measurement details are available in the supplementary material. On the same day as the imaging examinations, body fat and lean body mass were quantified using the dual energy X-ray absorptiometry (DXA) technique (GE Healthcare, Wauwatosa, WI, USA). Other clinical indices and scores were computed as discussed previously [8]. 

### 2.3. Imaging Biomarker Studies

The hepatic MRI-PDFF and MRE-LSM were measured with a 3-T scanner (MAGNETOM Skyra; Siemens Healthineers, Erlangen, Germany) using an 18-channel body matrix coil and table-mounted 32-channel spine matrix coil, as detailed previously [8]. As previously described, TE was conducted with FibroScan 502 (Echosens, Paris, France) by a certified technician who was blinded to the clinical and histological data [8]. All imaging studies were performed on the same day or within several days of blood sampling and just before bariatric surgery or liver biopsy (on the same day or several days before).

### 2.4. Liver Tissue Sampling and Analyses

Liver biopsy and tissue sampling were performed on 12 study participants in the cohort which included parent studies involving participants during bariatric surgery, donor liver resection for liver transplantation, and percutaneous liver biopsy procedures due to abnormal liver function. The liver tissues were analyzed as described previously [8]. The Nonalcoholic Steatohepatitis Clinical Research Network histologic scoring method was used to provide histological grading, which included NAS and fibrosis stages [23]. RNA sequencing (RNA-seq) analyses were performed on 12 liver samples from study subjects with a spectrum of NAFLD stages. Transcripts per million mapped reads were used for mRNA expression.

### 2.5. Metabolomics Analysis

For the metabolomics study using plasma samples from study subjects, profiling analyses of fatty acids (FAs), organic acids (OAs) and AAs were performed by GC-MS/MS as *tert*-butyldimethylsilyl (TBDMS)-, methoxime (MO)-TBDMS-, and ethoxylcarbonyl (EOC)-TBDMS derivatives, respectively, as previously described [24,25]. Profiling analyses of kynurenine pathway metabolites, nucleosides, and AAs were performed by LC-MS/MS using our platforms as detailed previously [24]. Further detailed descriptions of each analysis, sample preparation, and data processing are also available in the supplementary material.

### 2.6. Star Pattern Recognition Analysis

The levels of AAs, kynurenine pathway metabolites, nucleosides, OAs, and FAs were calculated using calibration curves. Finally, the mean values of each metabolite in the NAFL and NASH groups were adjusted to the corresponding mean values of the control group to normalize them. Each normalized value was represented by a line emanating from the same center point [26,27].

### 2.7. Identification of Potential Biomarkers and Related Metabolic Pathways

Partial least squares discriminant analysis (PLS-DA), a supervised machine learning tool, was performed for multivariate statistical analysis to distinguish the three groups and search biomarker candidates. The analysis was performed with log-transformed and autoscaled data. To identify the top-ranked altered pathways and depict significant biomarkers altered in NAFL and NASH, the metabolite set enrichment analysis and pathway analysis were conducted using Metaboanalyst (https://www.metaboanalyst.ca (accessed on 10 February 2021)).

### 2.8. Machine Learning and Multinomial Logistic Regression

Anonymized clinical data, including imaging and blood biomarker data, were collected along with metabolomics data from 89 participants. For comparison with our novel diagnostic index, we also calculated clinical scores such as Fibrosis-4 (FIB-4) and the NAFLD fibrosis score (NFS). Plasma metabolomics data comprised 79 metabolites, which included AAs, nucleosides, OAs, FAs, and others. To establish an ML model differentiating NASH, NAFL, and healthy control statuses, we executed a random forest (RF) algorithm with tuned optimal parameters through a 10-fold cross validation (CV) and Leave-One-Out Cross-Validation (LOOCV) method using the R package caret as described previously [28]. We randomly split the dataset into training and test sets with proportions of 75% and 25%, respectively. Thus, the datasets for ML to distinguish healthy and NAFL subjects were divided into 51 training sets and 16 test sets. The datasets of participants in the healthy and NASH groups were split into 34 training sets and 10 test sets, and data sets from the NAFL and NASH groups were divided into 47 training sets and 14 test sets. We excluded one outlier sample from a patient with NASH in aspartic acid data analysis (outlier value = 382.4 μg/μL) and the mean and median values of aspartic acid in 60 patients with NAFLD after the exclusion were 25.4 μg/μL and 16.9 μg/μL, respectively, before performing ML.

To evaluate the performance of the ML model, we determined the accuracy, kappa, and F1-score based on the AUROC curve and selected the top feature by comparing all values generated from different numbers of selected features (i.e., f79, f64, f32, f16, f8, and f4). We also assessed multinomial logistic regression (MLR) to build a best-fit model to predict the probability of NASH from RF-featured plasma metabolites. Additionally, we determined statistical significance using the computed coefficients and standard deviation (SD) with a two-tailed z test. To identify ML-featured plasma metabolites and their concentrations, we produced a decision tree using R.

To perform ML including data processing, MLR, and decision tree construction, we used RStudio (v4.0.2) with R (v 1.3.1073) containing R packages including *dplyr*, *stringr*, *reshape2*, *naniar*, *skimr*, *caret*, *pROC*, *multipleROC*, *MLmetrics*, *Nnet*, *Rpart*, and *Rattle*.

### 2.9. Multinomial Logistic Regression-Based Feature Selection

We also assessed multinomial logistic regression (MLR) to build a best-fit model to predict the probability of NASH from RF-featured plasma metabolites [29]. Additionally, we determined statistical significance using the computed coefficients and standard deviation (SD) with a two-tailed z test. To indicate NASH diagnostic capacity, green windows are displayed from the point where the probability of NAFL and NASH cross, as the probability of healthy control converges to 0 in the MLR plots.

### 2.10. Decision Tree-Based Feature Selection

To identify ML-featured plasma metabolites and their concentrations, we produced a decision tree (DT) using R package version 4.1–15. (https://CRAN.R-project.org/package=rpart (accessed on 7 June 2021)). In the DT, the method was set to “class” to compute the probability of each node forming a classification tree. The input data are then 8 RF-featured metabolites.

To perform ML including data processing, MLR, and DT construction, we used RStudio (v4.0.2) with R (v 1.3.1073) containing R packages including *dplyr*, *stringr*, *reshape2*, *naniar*, *skimr*, *caret*, *pROC*, *multipleROC*, *MLmetrics*, *Nnet*, *Rpart*, and *Rattle*.

### 2.11. Formulation of metaNASH Score

The score was calculated by assigning weights to the three important features obtained through the decision tree. There are two approaches: strengthening importance by giving high weight to features with high importance and balancing importance by giving high weight to features with low importance. We chose to balance the importance of features by reflecting the feature selection findings using ML and MLR.
$$Weight\;\left(w\right)=2^{\left(i-1\right)}\;\left(i=rank\right)\;f=the\;concentration\;of\;featured\;metabolites\;\left(\mathrm{mg}/L\right)\;Diagnostic\;index\left(score\right)=\overset{n}{\underset{i=1}{\prod }}f^{w}\;\left(n=the\;total\;number\;of\;features\right)\;MetaNASH=log_{10}\left({\left[Aspartic\;acid\right]}^{1}\times {\left[Isocitric\;acid\right]}^{2}\times {\left[Glutamic\;acid\right]}^{4}\right)\;$$

### 2.12. Statistical Analysis

Continuous variables are represented by the mean and standard deviation, while categorical variables are represented by the frequency (%). Statistical analyses of all datasets were performed and visualized with R. First, using the Shapiro Wilk test we evaluated the normality of all data to determine the appropriate statistical method. The Shapiro Wilk test indicated variations from the Gaussian distribution in some variables. Therefore, we chose to use nonparametric tests. Then, we applied either the Wilcoxon rank-sum test to examine two groups or the Kruskal-Wallis test to examine three groups using the R package *ggpubr* or Metaboanalyst. To overcome the multiple test problem, *p* values obtained from the Kruskal-Wallis test were adjusted by the false discovery rate (FDR) test.

The first and third quartiles of the data are indicated by the upper and bottom edges of the box, respectively, while the median is indicated by the center line. The whisker’s top and bottom edges represent the data’s maximum and minimum values. Visualization of metabolite and gene expression (heatmap), Spearman’s correlation (correlogram), and the gene network were assessed with the R packages including *ggpubr*, *ggplot2*, *igraph*, *corrr*, *corrplot*, *dplyr*, *tidyverse*, *ggraph*, *egg*, and *reshape2* as indicated previously [30,31]. All reported *p* values are two-sided and were considered statistically significant at <0.05.

## 3. Results

### 3.1. Characteristics of Participants

The demographics and clinical laboratory data obtained from the study cohort are summarized in Table 1. As anticipated, subjects with NAFLD showed significant difference compared to the control group in the majority of parameters.

### 3.2. Results of Metabolomics Analyses

A total of 79 metabolites (33 AAs, 4 kynurenine pathway metabolites, 4 nucleosides, 18 OAs, and 20 FAs) were detected by GC-MS/MS and LC-MS/MS in plasma samples from the healthy control, NAFL, and NASH groups. The plasma levels of the 79 metabolites are presented in Appendix A.

### 3.3. Plasma Metabolite Profiling and Univariate Analyses

In the NAFL group, five AAs (alanine, valine, glutamic acid, tyrosine, and α-aminoadipic acid), kynurenic acid, three OAs (2-hydroxybutyric acid, 3-hydroxypropionic acid, and α-ketoglutaric acid), and four FAs (myristoleic acid, palmitoleic acid, α-linolenic acid, and docosapentaenoic acid) were significantly increased compared to those in the healthy control group. However, hexacosanoic acid and glycine were significantly decreased in the NAFL group compared to the healthy control group (Appendix A).

In the NASH group, six AAs (alanine, valine, aspartic acid, glutamic acid, tyrosine, and 3-methylhistidine), kynurenic acid, 12 OAs (pyruvic acid, acetoacetic acid, glycolic acid, 2-hydroxybutyric acid, 3-hydroxybutyric acid, 3-hydroxypropionic acid, oxaloacetic acid, α-ketoglutaric acid, malic acid, 2-hydroxyglutaric acid, cis-aconitic acid, and isocitric acid), and four FAs (myristoleic acid, palmitoleic acid, γ-linolenic acid, and α-linolenic acid) were significantly increased compared to those in the healthy control group. However, 1-methylhistidine and erucic acid were significantly decreased in the NASH group compared to the healthy control group.

In the NASH group compared to the NAFL group, glutamic acid and three OAs (oxaloacetic acid, α-ketoglutaric acid, and isocitric acid) were significantly increased, while erucic acid was significantly decreased (Appendix A).

Among the metabolites, six AAs (alanine, valine, aspartic acid, glutamic acid, tyrosine, and α-aminoadipic acid), kynurenic acid, seven OAs (2-hydroxybutyric acid, 3-hydroxybutyric acid, 3-hydroxypropionic acid, oxaloacetic acid, α-ketoglutaric acid, malic acid, and cis-aconitic acid), and three FAs (myristoleic acid, palmitoleic acid, and α-linolenic acid) were significantly increased (*p* < 0.05), while glycine was significantly decreased (*p* < 0.05) in the NAFL and NASH groups compared to those of the control group. With an additional FDR test, we confirmed that six metabolites of two AAs (glutamic acid, and tyrosine), kynurenic acid, α-ketoglutaric acid, and two FAs (myristoleic acid, and palmitoleic acid) among these 18 metabolites were significantly different in the NAFL and NASH groups compared to those of the control group. These six metabolites are presented in Table 2.

### 3.4. Identification of Potential Biomarkers and the Construction of Metabolic Pathways

The metabolite profiles are presented as a heatmap in Figure 1A and as an enlarged heat map in Appendix A. In particular, six metabolites (glutamic acid, tyrosine, kynurenic acid, α-ketoglutaric acid, myristoleic acid, and palmitoleic acid) were significantly different between the three groups. Glutamic acid, tyrosine, α-ketoglutaric acid, myristoleic acid, and palmitoleic acid gradually increased according to disease progression from NAFL to NASH (Table 2 and Figure 1B). In supervised learning, PLS-DA was performed to identify biomarker candidates, and cross validation indicated that five metabolites were the most predictive of NAFLD progression. The PLS-DA score plot showed slight separation between three groups with a correlation coefficient (R2), accuracy, and cross-validation correlation coefficient (Q2) of 0.753, 0.533, and 0.341, respectively (Figure 1C). In the PLS-DA, the variable importance point (VIP) score was used for discrimination between the three groups (Figure 1D). Among the 79 metabolites, glutamic acid, myristoleic acid (C14:1), α-ketoglutaric acid, 3-hydroxypropionic acid, tyrosine, palmitoleic acid (C16:1), kynurenic acid, cis-aconitic acid, isocitric acid, malic acid, alanine, 3-hydroxybutyric acid, 2-hydroxybutyric acid, glycolic acid, acetoacetic acid, aspartic acid, docosanoic acid (C22:0), α-linolenic acid (α-C18:3), pyruvic acid, 2-hydroxyglutaric acid, γ-linolenic acid (γ-C18:3), valine, and oleic acid (C18:1) showed VIP scores >1.0, suggesting that they were major contributing metabolites for discrimination of the three groups.

Plasma levels of the 79 metabolites were subjected to pathway and enrichment analyses based on the library of Homo sapiens (Kyoto Encyclopedia of Genes and Genomes, KEGG), and the relevant metabolic pathways are shown in Figure 1E. Pathways with a calculated pathway impact value (>0.1) were considered potential target pathways. In the NAFL group compared to the healthy control group, pathways involving glutamine/glutamate metabolism, alanine/aspartate/glutamate metabolism, arginine/proline metabolism, arginine biosynthesis, and phenylalanine/tyrosine/tryptophan biosynthesis were selected as potential target pathways (Appendix A). In the NASH group compared to the healthy group, the glutamine/glutamate metabolism, alanine/aspartate/glutamate metabolism, arginine biosynthesis, the tricarboxylic acid (TCA) cycle, and arginine/proline metabolism pathways were selected as potential target pathways (Appendix A). In the NASH group against the NAFL group, glutamine/glutamate metabolism, the TCA cycle, arginine biosynthesis, alanine/aspartate/glutamate metabolism, and ketone body metabolism were selected as potential target pathways (Figure 1E).

### 3.5. Star Pattern Recognition Analysis

The levels of 79 metabolites in the NAFL and NASH groups were normalized to the corresponding mean levels of the control group. The resulting data are more informative when comparing the levels of altered metabolites (ranging from 0.62 to 2.84) to the control group. Therefore, the different star graphic patterns of the metabolites were expected to be useful for visual discrimination of the three groups. The normalized values of 33 AAs, 4 kynurenine pathway metabolites, 4 nucleosides, 18 OAs, and 20 FAs were used to draw star graphs with their respective numbers of rays. The differences between the NAFL, NASH, and control groups were shown as visual star patterns. In the tritriacontagon-shaped star pattern for AAs (Figure 2A), among 33 plasma AAs of the 79 metabolites, glutamic acid was the metabolite that was most increased in the NAFL and NASH groups—by 95% and 143%, respectively—compared to the control group. The 4-hydroxyproline level showed the most prominent decrease in plasma of the NAFL and NASH groups, decreasing by 33% and 38%, respectively, compared to the control group (Figure 2A). In the star pattern of octagonal shape for kynurenine pathway metabolites and nucleosides (Figure 2B), kynurenic acid was the metabolite that showed the most prominent increase in plasma of the NAFL and NASH groups, by 88% and 70%, respectively, compared to the control group. In the star pattern of octadecagonal shape for OAs (Figure 2C), among the 18 OAs, 16 OAs, excluding malonic acid and citric acid, were increased in the NAFL and NASH groups compared with the control group. In the star pattern of eicosagonal shape for the 20 FAs (Figure 2D), the myristoleic acid level was the most increased in the NAFL and NASH groups, increasing by 74% and 150%, respectively, compared to the control group.

### 3.6. Random Forest Algorithm-Based Modeling Predicts Metabolites That Distinguish the Progression of NAFLD

In addition to initial metabolomics data processing by routine statistical methods, we implemented ML methods to find metabolite-based biomarkers that distinguish patients with NASH, patients with NAFL and/or healthy subjects. First, we used the RF algorithm, which is an ensemble learning method that randomly generates multiple decision trees during training and is one of the most widely used algorithms for biomarker discovery [22,32]. We conducted RF modeling with a ten-fold cross validation method to establish several classifiers between NAFL vs. healthy controls, NASH vs. healthy controls, and NASH vs. NAFL. As detailed in the methods section, all datasets used for ML modeling were split randomly into 75% training and 25% test sets. To obtain the optimized classifier, we sequentially computed from the maximum of 79 combinations to the minimal combinations comprising four features (metabolites) (Figure 3A–C). The classifier of NAFL vs. healthy controls by combining eight features (f8) generated an AUROC curve of 0.900 (95% CI: 0.854, 0.947) (Figure 3A). In the discrimination of patients with NASH from healthy control subjects, the AUROC curve values of all computed classifiers were higher than 0.960, and the classifier containing four features provided the highest AUROC curve (0.990) (Figure 3B). However, the AUROC curve values of the NASH vs. NAFL classifier were relatively lower than those of other between-group classifiers, with the highest AUROC curve of 0.849 with features comprising four metabolites (Figure 3C). To predict optimal classifiers and assess the performance of the RF models in healthy control-NAFL-NASH discrimination, we also computed the accuracy (blue curve), F1-score (green curve), AUROC curve value, and kappa value (purple curve in the separated plots) of each classifier (Figure 3D). The overall accuracies and F1 scores displayed similar trends to the AUROC curve values. However, in distinguishing NASH and NAFL, the accuracy and F1-score decreased slightly while the AUROC curve value increased as the number of features in the model increased. After considering accuracies, F1-scores, AUROC curve values, and kappa values, eight metabolites (f8) for NAFL vs. healthy controls, four metabolites (f4) for NASH vs. healthy controls, and eight metabolites (f8) for NASH vs. NAFL discrimination were selected as the representative classifiers. The metabolites in each classifier are presented as boxplots in Figure 3E–G. Only glutamic acid was a common classifier in the three comparative analyses. Among the NAFL versus healthy control classifier metabolites (f8), four metabolites (α-aminoadipic acid, glutamic acid, glycine, and myristoleic acid) were significantly different between the NAFL group and the control group, but the other four metabolites were comparable between the two groups, although ML modeling selected all eight metabolites (Figure 3E). Among the featured metabolites, glutamic acid, α-ketoglutaric acid, myristoleic acid, and tyrosine were increased significantly in the NASH group compared to the healthy control group (Figure 3F). Glutamic acid, isocitric acid, α-ketoglutaric acid, and oxaloacetic acid were higher in the NASH group than in the NAFL group, while plasma levels of cis-aconitic acid, aspartic acid, 2-hyroxyota decanoic acid, and 4-hydroxphynyl lactic acid were comparable between the two groups (Figure 3G). Collectively, the RF models for each binary group format featured fourteen metabolites, among which only six metabolites (i.e., glutamic acid, isocitric acid, α-ketoglutaric acid, oxaloacetic acid, myristoleic acid, and tyrosine) were significantly elevated in the NASH group compared to either the healthy control or NAFL group. We further performed modeling with the LOOCV method to distinguish between patients with NASH and NAFL (Appendix A). It consistently proposed the same features that the ten-fold CV selected.

### 3.7. Multinomial Logistic Regression Analysis Identified Eight Plasma Metabolites

To further optimize the fourteen RF-featured metabolites, we conducted multinomial logistic regression (MLR). First, we calculated the probability of eight metabolites from RF modeling predicting the three conditions according to their plasma concentrations and clinical parameters (MRI-PDFF, C3, NFS, and FIB-4) (Figure 4A,B). The probabilities of the prediction of NAFL and NASH versus healthy controls, respectively, were also estimated (Appendix A). The probability of the MLR models was visualized with each line-plot showing three curves for the healthy control (gray), NAFL (red), and NASH (blue) groups. The green window in each line plot presents the range with accurate NASH diagnostic capability. The green windows of the MLR plots indicate the variable diagnostic performance (the width of the green window) in discriminating NASH from NAFL or healthy control states. Six metabolites, cis-aconitic acid, aspartic acid, glutamic acid, isocitric acid, α-ketoglutaric acid, and oxaloacetic acid, corresponded to green windows among the eight RF-predicted metabolites for discriminating NASH and NAFL states, which was similar to other diagnostic indices. Only glutamic acid and myristoleic acid generated green windows with acceptable widths among the eight RF-predicted metabolites separating NAFL from the healthy control state (Appendix A), while four RF-predicted metabolites (glutamic acid, α-ketoglutaric acid, myristoleic acid, and tyrosine) performed well in distinguishing NASH from the healthy control state in MLR modeling (Appendix A). Notably, increased plasma levels of 2-hydroxybutyric acid and 4-hydroxyphenyl lactic acid were associated with a higher probability of NAFL (red curve) than NASH (blue curve) across the detectable ranges (Figure 4A). Together, the eight metabolites (cis-aconitic acid, aspartic acid, glutamic acid, isocitric acid, α-ketoglutaric acid, oxaloacetic acid, myristoleic acid, and tyrosine) generated acceptable green windows, in which the MLR probability of NASH was higher than that of both healthy individuals and patients with NAFL, revealing the diagnostic power of the fourteen RF-featured metabolites. The NASH group presented higher concentrations of all eight metabolites than the healthy control group, while the subjects with NASH had significantly higher plasma concentrations of four metabolites (glutamic acid, isocitric acid, α-ketoglutaric acid, and oxaloacetic acid) than patients with NAFL (Figure 4C). Only two metabolites (glutamic acid and α-ketoglutaric acid) showed consistent and gradual increases in plasma according to NAFLD progression. In a parallel MLR analysis, we also calculated the probability of four clinical parameters predicting NASH versus NAFL (Figure 4B). We have used the four parameters applied in the assessment of NAFLD to reflect the progression of steatosis (MRI-PDFF), inflammation (C3), and fibrosis (NFS and FIB-4) [6,33].

### 3.8. Decision Tree Algorithm Defined the Three Most Critical Plasma Metabolites, Distinguishing Patients with NAFL from Those with NASH

In terms of good diagnostic indices, not only their sensitivity and specificity but also their economy and simplicity are critical. To achieve this goal, we further optimized the eight MLR-selected metabolites, which are presented in Figure 4C, by recursive partitioning and regression tree (RPRT) methods to predict NASH. The input data were eight RF-featured metabolites (cis-aconitic acid, aspartic acid, glutamic acid, 2-hydroxybutyric acid, 4-hydroxyphenyllactic acid, isocitric acid, α-ketoglutaric acid, and oxaloacetic acid) (Figure 3G and Figure 4A). The optimized decision tree distinguishing NASH from NAFL indicated that three metabolites (aspartic acid, isocitric acid, and glutamic acid) were sufficient to discriminate patients with NASH from control subjects or patients with NAFL (Figure 5A). 

### 3.9. MetaNASH Score: A Metabolite-Based NASH Diagnostic Tool with Acceptable Performance

We generated the following formula by assigning weights from the bottom of the optimized decision tree and including the three metabolites as we described in the method Section 2.11 and named it the “*MetaNASH* score” (Figure 5B). Among the approaches presented in the method Section 2.11, the AUROC obtained by giving high weight to features with high importance was 0.719, and the AUROC obtained by giving high weight to features with low importance was 0.821. Aspartic acid was, in fact, the feature with the highest importance obtained through the decision tree. The boxplot analysis revealed no significant difference in the concentration of aspartic acid between the NAFL and NASH groups (Figure 3G). Thus, we believe that the formula for computing the diagnostic index using the approach of balancing importance is quite reasonable.
$$MetaNASH=log_{10}\left({\left[Aspartic\;acid\right]}^{1}\times {\left[Isocitric\;acid\right]}^{2}\times {\left[Glutamic\;acid\right]}^{4}\right)$$

In our cohort, *MetaNASH* scores ranged from 1.78 to 7.79. The mean (SD) values in the three experimental groups were 3.20 (0.84) in the healthy group, 4.44 (1.10) in the NAFL group, and 5.25 (0.72) in the NASH group. Since the optimal value proposed by AUROC analysis for models including multiple predictors should be applicable in clinical practice, our strategy was to minimize the chance of false negatives. For this purpose, the point with the highest specificity was selected as the optimal value among cases with the highest sensitivity in multiple modeling, which was the same point obtained by Youden’s J index method [34]. Thus, the optimal cutoff value of 4.543 (AUROC = 0.821, F1 score = 0.746 and kappa = 0.478) for the *MetaNASH* score achieved 72% accuracy. At least in our cohort, the *MetaNASH* score with this cutoff level did not miss any NASH cases (Figure 5B).

We compared the *MetaNASH* score system to other parameters (i.e., C3, NFS, and FIB-4). The green dotted line of each scatter plot displays the cutoff value of each diagnostic index for NASH in our own algorithm: 4.543 for the *MetaNASH* score, 175 for the serum C3 level, and −1.46 for the NFS (Figure 5C) [6]. The *MetaNASH* score, C3, and FIB-4 classified 2, 0, and 1 healthy participants as patients with NASH, while the NFS classified 22 healthy control subjects as patients with NASH. Among 19 patients with NASH, the *MetaNASH* score successfully identified all cases, whereas C3, NFS, and FIB-4 correctly identified 5, 16, and 7 cases, respectively. Next, we compared the AUROC curve, accuracy, F1-score, and kappa values of all diagnostic indices. The *MetaNASH* score had better or comparable performance and reliability compared to the other parameters (Figure 5D).

Since BMI, insulin, and glucose levels were significantly altered in patients with NASH compared to patients with NAFL (Table 1), we investigated whether BMI, glucose, and insulin levels impact the *MetaNASH* score (Table 3). After resampling NAFL and NASH patients with similar BMI, insulin, and glucose levels, the performance of the *MetaNASH* score was tested. With a *MetaNASH* score of 4.543, it was possible to distinguish patients with NASH from patients with NAFL.

### 3.10. The MetaNASH Score Performed Better Than the GSG Index and Glutamic Acid/Glutamine Ratio in the Discrimination of NASH

The glutamate–serine–glycine (GSG) index (glutamate/(serine + glycine)) [35] and glutamate/glutamine ratio [36] were reported to distinguish NASH from non-NASH. Thus, we compared the *MetaNASH* score with the GSG index and glutamic acid/glutamine ratio in our whole cohort without separation into training and test sets. As shown in Figure 6, both the GSG score and the glutamic acid/glutamine ratio were comparable between the NAFL and NASH groups, and their performances in the discrimination of NASH were also lower than that of the *MetaNASH*. However, the *MetaNASH* score was higher in the NASH group than in the control and NAFL groups. When we reassess the performance of the *MetaNASH* score in the whole cohort without splitting into training and test sets, the AUROC curve of the *MetaNASH* score for the discrimination of NASH was 0.877, with a cutoff value of 4.55.

### 3.11. Metabolic Remodeling in NASH Was Associated with Altered Gene Expression Profiles in the Liver

To expand our understanding of the association between NASH and eight ML-featured plasma metabolites, we performed multi-omics analysis using metabolomics and transcriptome datasets. First, we summarized the plasma metabolomics using eight ML-selected features. As expected, the plasma levels of the eight ML-featured metabolites increased gradually from the healthy to NAFL to NASH states (Figure 7A). Then, we analyzed a publicly available dataset from RNA-seq analysis of 10 healthy control subjects, 51 patients with NAFL, and 155 patients with NASH, which was generated by Govaere et al. (NCBI GEO GSE135251) [37]. After obtaining the whole expression profiles from the RNA-seq data, we selected and summarized the expression profiles of genes that directly govern the biochemical pathways of the eight ML-featured metabolites (i.e., ML-featured metabolite-associated genes, MAGs). The expression of all MAGs, which encode enzymes directly catalyzing reactions involving the eight metabolites, was altered in NASH livers. Eleven genes (i.e., *DLD*, *IDH2*, *CS*, *DLST*, *MDH2*, *ACO2*, *IDH3G*, *OGDH*, *PCK2*, *ACO1* and *PC*) were increased, while nine genes (i.e., *IDH3A*, *MDH1*, *MDH1B*, *GOT2*, *PCK1*, *GLUD1*, *GOT1*, *IDH1* and *TAT*) were reduced in the NASH state (Figure 7B). In particular, the expression of genes involved in de novo lipogenesis (DNL) was significantly increased in NAFL and NASH except for ACACA (Figure 7C). However, the trend of hepatic gene expression in relation to fatty acid β-oxidation (FAO) was not homogenous. The expression of only 6 genes (*ACAA2, ACADL, ACADM, CPT1A, ACAT1,* and *HADHB*) among 16 genes that we selected was reduced in the NASH state, while the remaining 10 genes showed increasing patterns in NAFL and NASH. We constructed an integrated metabolic map of NAFLD based on the metabolomics results from the present study (Figure 7D). Although this metabolic map summarized metabolic remodeling during NAFLD progression into the same dimension using information from the circulation (metabolome) and liver (transcriptome), the abundances of plasma metabolites and their related hepatic transcripts were significantly correlated and reflected disease progression in the liver. As shown in the metabolic map, the majority of the featured genes and metabolites were elevated in both NAFL and NASH. Interestingly, in the NASH liver, most genes governing mitochondrial metabolism were generally upregulated, whereas genes regulating metabolism in the cytoplasm were downregulated. To identify genes specifically altered in NASH, we also performed RNA-seq on liver tissues from patients with NAFLD (*n* = 12). Then, we classified the patients into two groups based on the NAS (NAS < 4 or ≥ 4). Among 20 genes belonging to MAGs, 16 genes were elevated in patients with NAS ≥ 4 (Figure 7E, top). In patients with NAS ≥ 4 compared to the patients with NAS < 4, three DNL genes were upregulated, while nine FAO genes were downregulated (Figure 7E, bottom). To further identify NASH-specific altered genes, we assessed commonly altered genes related to the NASH state between our RNA-seq data and the dataset of Govaere et al. (NCBI GEO GSE135251) [37]. A total of 7 genes among the 20 MAGs, 3 DNL genes, and 9 FAO genes were altered by the same patterns in both datasets (Figure 7F), implying that these genes are closely associated with NASH. In addition, two network analyses visualizing coexpression (based on Spearman’s Rho) in the non-NASH (NAS < 4) and NASH (NAS ≥ 4) states showed their own remarkable association patterns (Figure 7G,H). In the non-NASH state, the MAGs, DNL and FAO genes were tightly associated and expressed together with positive (blue edges) or negative (red edges) correlations. However, the total number of edges reflecting coexpression was reduced in the livers of patients with NASH. New patterns of strong positive or negative associations were generated, which implied that dysregulation of these metabolic genes in and around mitochondria could be a dominant driver of specifically altered metabolism in patients with NASH.

## 4. Discussion

Noninvasive accurate biomarkers for the diagnosis of NASH/fibrosis and the identification of NAFLD progression are urgently needed, and these biomarkers should also be measurable in serial assessments of the same patients with NAFLD with or without comorbidities. ML has been introduced to analyze metabolomics data and has provided impressive predictive competencies [22]. These ML tools apply distinct strategies for statistical analyses to yield the best performance in the identification of NASH from non-NASH.

In the present study, we performed plasma metabolomics analysis in healthy control subjects and patients with NAFLD to obtain a metabolic signature that reflects the progression of NAFLD and distinguishes NASH from non-NASH. We determined 79 plasma metabolites, including AAs, kynurenine pathway metabolites, nucleosides, OAs, and FAs, in the study subjects using GC-MS/MS and LC-MS/MS. Six metabolites (glutamic acid, tyrosine, kynurenic acid, α-ketoglutaric acid, myristoleic acid, and palmitoleic acid) were significantly different between the three groups, with five metabolites, except for kynurenic acid, increasing gradually according to progression from control to NAFL to NASH. However, these five plasma metabolites were suboptimal for the discrimination of NASH from NAFL or controls. Thus, by applying multiple ML methods to the interpretation of plasma metabolomics data, we tried to define a set of metabolite biomarkers that better distinguished NASH from NAFL and healthy controls. RF and MLR analyses showed that eight metabolites (glutamic acid, cis-aconitic acid, aspartic acid, isocitric acid, α-ketoglutaric acid, oxaloacetic acid, myristoleic acid, and tyrosine) could distinguish the three groups. The RPRT algorithm led to the final set of three metabolites (glutamic acid, isocitric acid, and aspartic acid). With the three selected metabolites, we developed an excellent metabolomics index for the discrimination of NASH, the *MetaNASH* score, which gave glutamic acid a weight corresponding to the fourth power of the plasma concentration and assigned a weight corresponding to the second power of the plasma level to isocitric acid. The *MetaNASH* score had an AUROC curve value of 0.877 at a cutoff value of 4.55 in the whole cohort analysis. In addition, we integrated our metabolome data into RNA-seq datasets and constructed a map of metabolism in relation to NASH, which showed specific and altered patterns of associations between the concentration of plasma metabolites and hepatic gene expression of enzymes governing energy metabolism including the TCA cycle, DNL, and mitochondrial metabolism.

Although metabolic stimulation of hepatocytes can increase the expression of AST and ALT genes, the mitochondrial isotype protein expression and intracellular activities of the two enzymes were reported to be decreased, whereas extracellular release increased with stimulation [18]. Circulating levels of AST, ALT, and gamma glutamyl transferase (GGT) do not accurately reflect the presence and progression of steatosis or NASH [18,35,38]. Thus, it would be reasonable to determine whether there are any measurable metabolites that better reflect hepatic responses according to NAFLD progression.

NAFLD, diabetes, insulin resistance (IR), and obesity usually share common pathophysiology. Thus, circulating levels of BCAAs, aromatic AAs, and other AAs, such as glutamate, alanine, and aspartate, have been shown to be increased under these conditions [18,35,36,39,40], while glycine and serine were found to be decreased [35,39]. In the present study, glutamic acid and α-ketoglutaric acid showed consistent and gradual increases in plasma according to NAFLD progression in both statistical analysis and ML-based modeling analyses. The mean plasma concentrations (±SDs) of glutamine acid in the control, NAFL, and NASH groups were 6.52 (2.47) μg/mL, 12.74 (6.45) μg/mL, and 15.88 (6.36) μg/mL, respectively, while those of α-ketoglutaric acid were 1.70 (0.56) μg/mL, 2.63 (1.32) μg/mL, and 3.71 (1.85) μg/mL, respectively. The increase in glutamic acid alongside other AAs suggested a higher rate of whole-body protein turnover, increased transamination of AAs, and increased oxidative stress with the progression of NAFLD [15]. Plasma α-ketoglutarate has been shown to be increased in obesity and NAFLD [21]. Metabolic imbalance with mitochondrial dysfunction in these conditions can cause an increase in plasma α-ketoglutarate levels from the early stage of disease, probably representing a compensatory mechanism [21,41].

Glutamate is involved in many transaminase reactions, including ALT, AST, and GGT, in the synthesis of glutathione, and in the urea cycle and related metabolite flow pathways [18,36,42,43]. In the study of a genome-wide metabolic model for hepatocytes, glutamate was the most connected node in the appearance of NASH [16]. In addition, glutaminolysis is a critical pathway of metabolic reprogramming for actively remodeling tissues in that glutamine via glutamate and α-ketoglutarate conversion contributes to TCA cycle activity to power cells [36,44]. Gaggini et al. [35] analyzed AA profiles in addition to the measurement of euglycemic insulin clamp IR indices in subjects with NAFLD without diabetes and healthy control subjects and developed a new glutamate–serine–glycine (GSG) index (glutamate/(serine + glycine)). In their report, the GSG index correlated with hepatic IR, mildly with HOMA-IR, but not with peripheral IR and was associated with ballooning and/or inflammation in liver biopsy and was able to discriminate fibrosis F3-4 from F0-2 in the examined cohort [35]. These three AAs were also reported to be altered with NASH progression in the study involving genome-scale metabolic model for hepatocytes [16]. However, this index was not validated in a recent study, in which the glutamate/glutamine ratio correlated with NASH [36]. The study performed by Du K et al. [36] showed that hepatic stellated cell (HSC) glutamine metabolism is also important for NASH and fibrosis progression in NAFLD. They showed a decrease in glutamine and an increase in the glutamate/glutamine ratio in serum from animal models of CCl_4_-induced fibrosis and diet-induced NASH and from patients with NASH [36]. Furthermore, a recent study showed that glutamate also directly acted on HSCs via metabotropic glutamate receptor 5, which led to increased production of 2-arachidonoylglycerol (2AG) [45]. Additionally, 2AG can stimulate lipogenesis and fibrosis via cannabinoid receptor 1 [45,46]. A metabolic flux study integrated with transcriptomic data, in which metabolites were measured across the human splanchnic vascular bed in patients with NAFLD, showed net uptake of cysteine, glutamine, serine, proline, threonine, tyrosine, alanine, glycine, ornithine, and others into the liver and net export of palmitate, triglyceride, cholesterol, and glutamate from the liver in both basal and euglycemic hyperinsulinemic states [39]. Thus, the abovementioned studies and our results indicate that glutamate is a concrete metabolite biomarker that reflects a decreased metabolic adaptability in NAFLD.

When we calculated the GSG index and glutamic acid/glutamine ratio in our cohort, their performance in differentiating NASH from non-NASH was suboptimal, in contrast to previous reports [35,36]. Our *MetaNASH* score showed excellent performance in the discrimination of NASH in the present study. These findings suggest that the study design and study subjects might also affect metabolite profiles. Our *MetaNASH* formula also included weighted isocitrate and non-weighted aspartic acid. Isocitrate is an intermediate metabolite involved in the glutaminolytic reaction, citrate export, transformation to α-ketoglutarate, and NADPH shuttling during metabolic stress and cell proliferation [47]. In addition, isocitrate dehydrogenase (IDH) is a key rate-limiting enzyme in the TCA cycle, which seems to be in line with the changes in hepatic IDH gene expression and the increase in plasma isocitrate in patients with NASH in the present study. Citrate/isocitrate may be an important player in NAFLD in that citrate promotes oxidative stress, acting as the strongest stimulator in the presence of iron via the Fenton reaction [48]. Thus, in our dataset and public transcriptome datasets, genes governing citrate/isocitrate/α-ketoglutarate pathways showed common changing patterns in NASH, supporting the feasibility of isocitrate as a biomarker. Aspartic acid is closely related to AA metabolism, which is initiated by aminotransferases with glutamate and α-ketoglutarate as reaction partners. AST transfers the glutamate amino group to oxaloacetate to produce aspartate [42], which suggests that aspartate is also a large node.

In the present study, metabolite profiles from the ML algorithms included metabolites that fuel the TCA cycle or are exported from the TCA cycle for lipogenesis and other pathways. However, in contrast to the literature [35,49,50], our results showed that circulating BCAAs were not associated with NAFLD progression in the present study, indicating that elevated BCAAs are rather metabolic features related to peripheral IR, particularly in muscle [35]. Decreased BCAA catabolism in muscle and adipose tissue can increase circulating BCAA levels, and the increased BCAAs can be used as substrates for lipogenesis in zone 3 hepatocytes [51]. In line with this view, it was reported in a splanchnic metabolic flux study that insulin increased the net uptake of BCAAs in the liver [39]. We observed that tyrosine is increased with the progression of NAFLD. Increased tyrosine was relatively consistently reported and shown to be correlated with systemic and hepatic IR [35]. These findings suggest that extrahepatic factors, such as sarcopenia, adiposity, and IR, also need to be considered when evaluating metabolomics data in NAFLD. In our cohort, the study subjects were relatively young and many patients with NAFLD were planning to receive bariatric surgery.

In multiomics analyses, we observed that ML-featured MAGs were altered in NASH livers, with a significantly increased expression of genes involved in DNL. The patterns of the expression of FAO-related genes were not homogenous, but with an increase in plasma isocitrate and changes in other gene expression patterns, suggested decreased FAO in the mitochondria in NASH. Interestingly, the abundances of plasma metabolites and their related hepatic transcripts were significantly correlated, and the genes of MAGs, DNL and FAO were tightly associated and positively or negatively correlated with each other. We observed that these correlation patterns were altered characteristically in NASH, implying dysregulation of these metabolic genes in and around mitochondria. Taken together, circulating metabolite profiles could reflect specific information originating from the liver harboring the altered gene expression profiles observed in this study.

One of the strengths of the present study is that we measured a broad spectrum of metabolites and analyzed the metabolomics results with multiple ML-based algorithms. Thus, we could profile eight metabolites with discriminative features for NASH. Second, we developed an innovative metabolomics index to distinguish NASH with excellent performance. The study limitations also need to be addressed. First, the majority of study subjects were not classified according to the biopsy results, but were classified according to our algorithm based on the combination of clinical data and imaging biomarkers in the majority of cases. However, the NASH definition in the present study definitely indicates a progressive form of NAFLD, considering the MRE cutoff value of 3.8 kPa [8], suggesting that the *MetaNASH* score can at least identify a progressive form of NAFLD. Second, we focused on the metabolites selected in ML-based algorithms and constructed a metabolic map based on the selected metabolites. Other metabolites might be selected upon variations in the study design, study population, and sample size [16,35,36].

In conclusion, our ML-based algorithm might capture the metabolic features of NAFLD progression well, and our *MetaNASH* score optimally distinguished the progression of NAFLD. Further study on the metabolites and their associated pathways in relation to NASH progression and validation of the *MetaNASH* score through independent cohort studies are warranted.

## Figures and Tables

**Figure 1 biomedicines-10-01669-f001:**
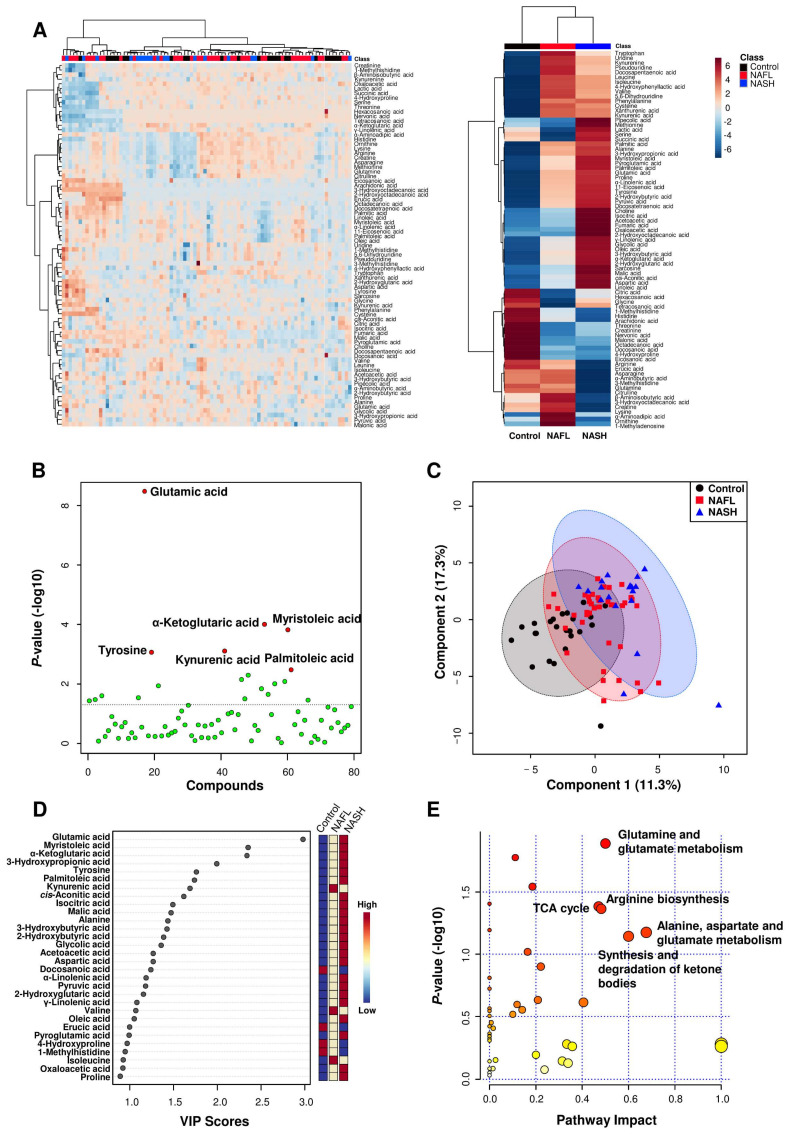
Bioinformatics analyses on plasma metabolite data in the healthy, NAFL, and NASH groups. (**A**) Hierarchical clustering heatmaps showing the normalized plasma metabolite levels (Z scores) of all participants (left) and mean levels of Z scores in each study group (right). (**B**) Scatter plot of the Kruskal-Wallis test. (**C**) PLS-DA score plot of the study groups. (**D**) Variable importance in projection (VIP) plot. (**E**) Bubble plots of altered metabolic pathways related to changes in plasma metabolites in the NASH group versus the NAFL group.

**Figure 2 biomedicines-10-01669-f002:**
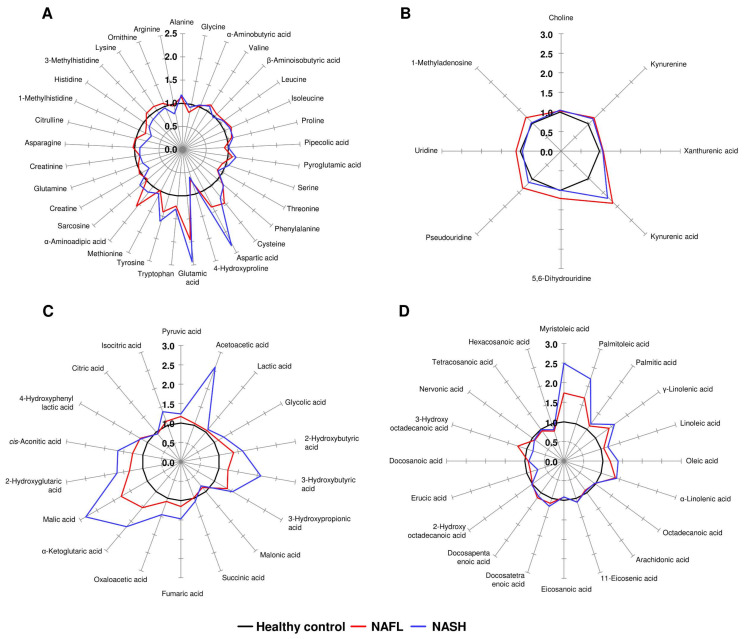
Star symbol plots of the study groups. The plots were drawn using the mean values of the study groups for 33 AAs (**A**), 4 kynurenine pathway metabolites and 4 nucleosides (**B**), 18 OAs (**C**), and 20 FAs (**D**) after normalization to the corresponding mean value of each metabolite in the control group.

**Figure 3 biomedicines-10-01669-f003:**
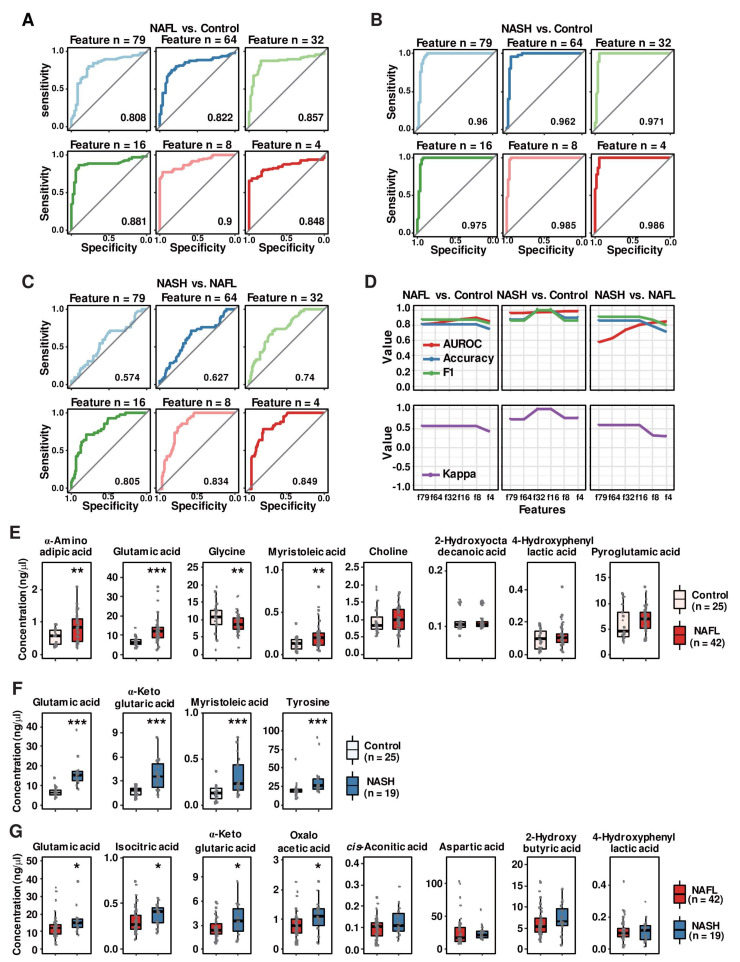
Predictive power of RF classification models and the comparison of metabolite abundances of RF-featured metabolites between groups. (**A**–**C**) AUROC curves of RF classification models for distinguishing the study groups: (**A**) NAFL vs. control; (**B**) NASH vs. control; and (**C**) NASH vs. control. (**D**) The predictive power of the classification models evaluated by AUROC curves, accuracies, F1-scores, and kappa values. (**E**–**G**) Boxplots showing the abundance of RF-featured classifier metabolites: (**E**) NAFL vs. control classifiers; (**F**) NASH vs. control classifiers; and (**G**) NASH vs. NAFL classifiers. Each scatter dot in the boxplots represents the concentration of each subject. The Wilcoxon rank sum test was used to determine significant differences between two groups. *, *p* < 0.05; **, *p* < 0.01; and ***, *p* < 0.001 between groups.

**Figure 4 biomedicines-10-01669-f004:**
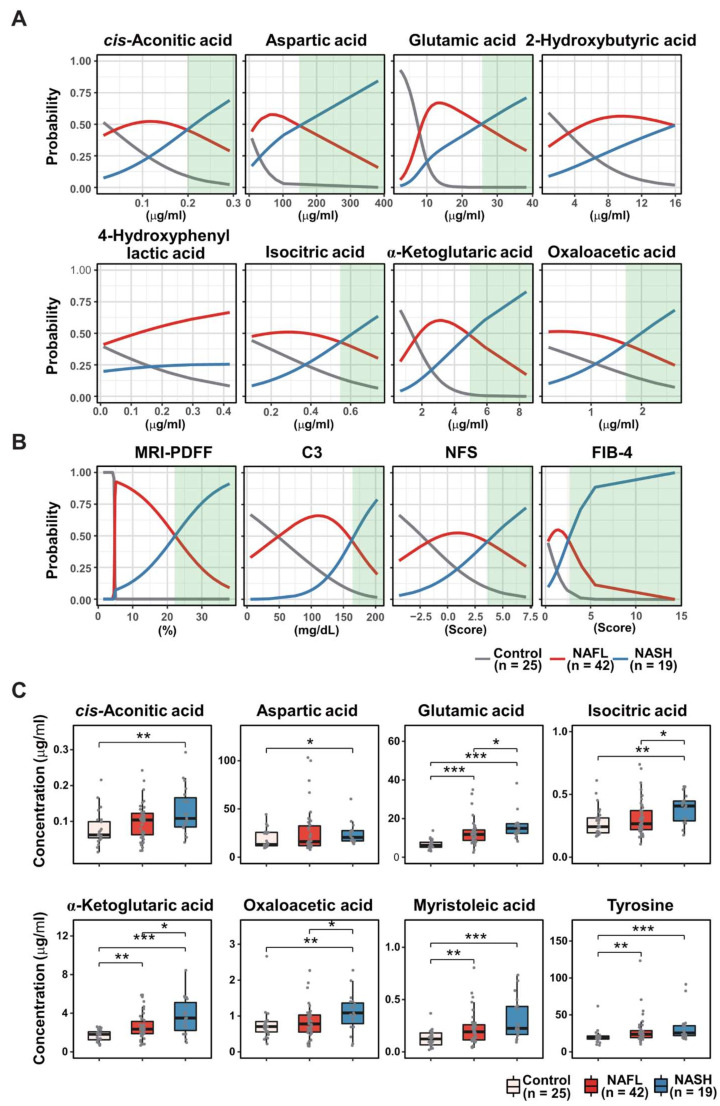
Multinomial logistic regression (MLR) estimating the performance of RF-featured metabolites and clinical parameters, including MRD-PDFF, C3, NFS, and FIB-4 in the discrimination of the three groups. MLR curves showing the probability of eight RF-featured metabolites (**A**) and of clinical parameters (**B**) discriminating NASH vs. NAFL. Green windows indicate the zone distinguishing NASH from non-NASH. (**C**) Boxplots showing the abundance of eight MRL-selected metabolites in the control, NAFL, and NASH groups. Significance was evaluated by the Kruskal-Wallis test. ***, *p* < 0.001; **, *p* < 0.01; *, *p* < 0.05.

**Figure 5 biomedicines-10-01669-f005:**
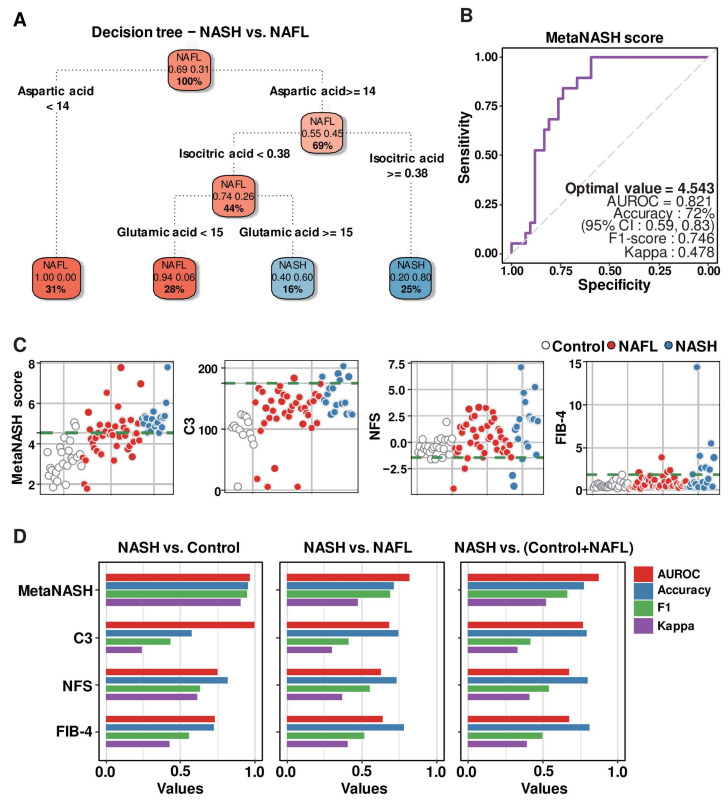
Development of the *MetaNASH* score as a diagnostic index for NASH. (**A**) Decision trees summarizing the process of ML-based selection of three essential features identifying patients with NASH. (**B**) AUROC curve highlighting the performance of the *MetaNASH* score. (**C**,**D**) Comparison of the predictive power of the *MetaNASH* score and other clinically applicable parameters. Scatter (**C**) and bar (**D**) plots comparing the predictive power of each diagnostic factor.

**Figure 6 biomedicines-10-01669-f006:**
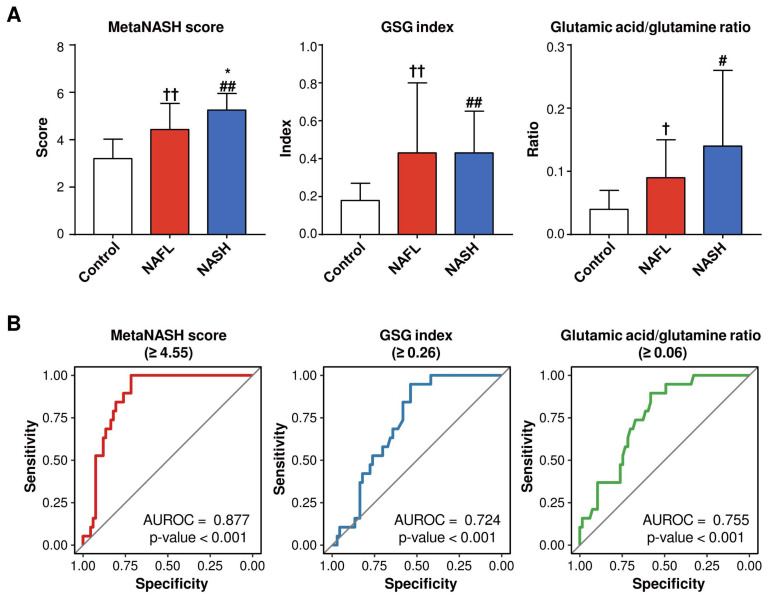
Comparisons of the *MetaNASH* score, the GSG index, and the glutamic acid/glutamine ratio between the study groups and their performance in the discrimination of NASH. (**A**) *MetaNASH* scores, GSG index values, and glutamic acid/glutamine ratios in the study groups. Data are expressed as the mean ± SD. The statistical significance of each variable of the NASH group compared to the control and NAFL groups was evaluated by post hoc analysis. ^†^, *p* < 0.05; and ^††^, *p* < 0.01 for NAFL vs. control: ^#^, *p* < 0.05; and ^##^, *p* < 0.01 for NASH vs. control: *, *p* < 0.05 for NASH vs. NAFL. (**B**) Performance of the *MetaNASH* score, the GSG index, and the glutamic acid/glutamine ratio in distinguishing NASH in the whole cohort.

**Figure 7 biomedicines-10-01669-f007:**
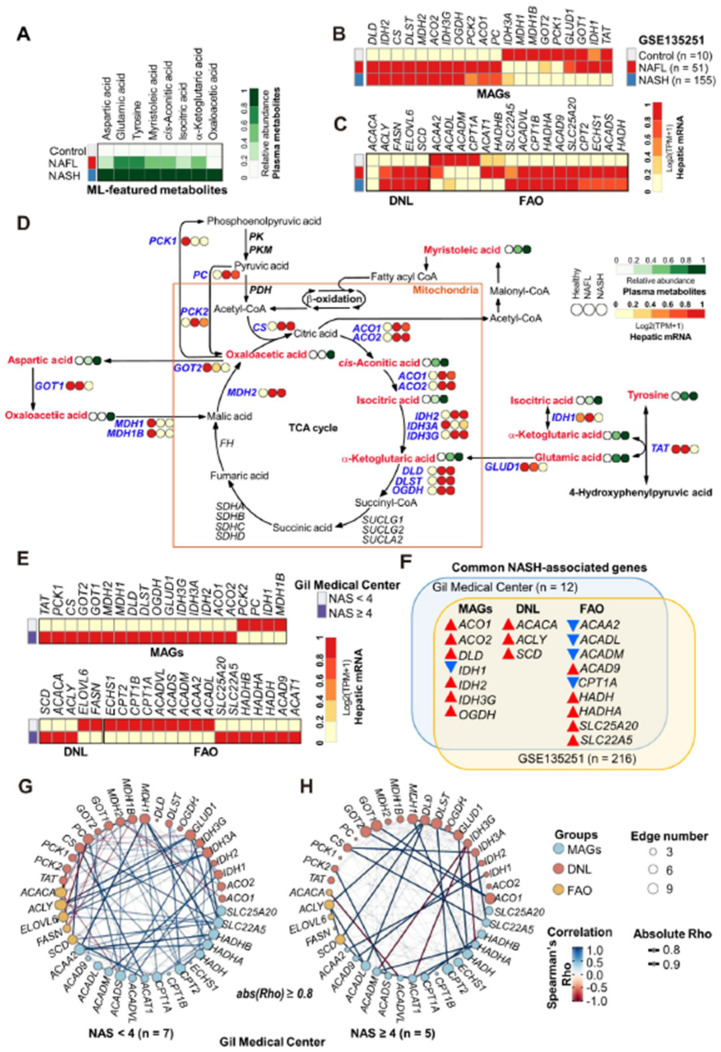
Multi-omics analyses presenting NASH-specific associations between ML-featured metabolites and their metabolic pathway-related genes. (**A**) Heatmap presenting the relative abundance of 8 ML-featured metabolites. (**B**) Heatmaps visualizing the relative expression of genes encoding enzymes governing metabolic pathways of ML-featured metabolites (MAGs). (**C**) Heatmaps visualizing the relative expression of genes involved in hepatic DNL and FAO. The gene expression profiles presented in (**B**,**C**) are from the RNA-seq dataset (GSE135251) described in the “Results” section. (**D**) Integrative metabolic map summarizing the associations of ML-featured metabolites and their associated genes with NAFLD progression. (**E**) Heatmaps displaying the expression levels of hepatic transcripts of MAGs, DNL, and FAO in our own RNA-seq data according to NAFLD progression (NAS < 4 (*n* = 7) versus NAS ≥ 4 (*n* = 5). (**F**) Venn diagram summarizing commonly altered genes between the two datasets (our dataset and the public dataset). Red and blue arrows indicate up- and down-regulated gene expression in subjects with NASH compared to subjects without NASH, respectively. (**G**) Coexpression analysis in subjects with NAS < 4 (non-NASH) and (**H**) NAS ≥ 4 (NASH) based on Spearman’s correlation. Correlations with an absolute Rho above 0.8 or more are only visualized as edges. A blue edge indicates Rho ≥ 0.8, and a red edge indicates Rho ≤ −0.8 with statistical significance.

**Table 1 biomedicines-10-01669-t001:** Demographic and clinical characteristics of the study subjects.

Characteristics	Control (*n* = 25)	NAFL (*n* = 42)	NASH (*n* = 19)	*p*-Values
**Age (years)**	35.2 (15.3)	43.2 (15.7) ^†^	41 (16.2)	0.11
**Sex (male/female)**	15/10	19/23	12/7	NA
**Weight (kg)**	65.9 (10.2)	84.4 (18.5) ^†††^	104.3 (29.4) ^###,^ **	<0.001
**BMI (kg/m^2^)**	23.2 (2.9)	30.6 (5.9) ^†††^	35.5 (7) ^###,^ **	<0.001
**Waist circumference (cm)**	79.9 (6.8)	100.6 (14.7) ^†††^	113.5 (15.8) ^###,^ **	<0.001
**SBP (mmHg)**	128.7 (18.3)	128.6 (14.4)	130.2 (14.1)	0.93
**DBP (mmHg)**	82.8 (12)	85.2 (9.4)	87.2 (12.3)	0.3
**AST (U/L)**	20.4 (5.5)	42.8 (45.1) ^†††^	87.1 (58.1) ^###,^ ***	<0.001
**ALT (U/L)**	18.4 (7.4)	62.3 (78.3) ^†††^	118.8 (100.3) ^###,^ *	<0.001
**GGT (U/L)**	18.3 (8.2)	45.7 (32.1) ^†††^	96.7 (58.8) ^###,^ ***	<0.001
**Total cholesterol (mg/dL)**	189.3 (33.9)	194.5 (35.6)	180.9 (38.6)	0.4
**HDL-C (mg/dL)**	59.3 (14.2)	50.5 (11.2) ^††^	49.4 (27.2) ^##^	<0.05
**Triglycerides (mg/dL)**	97.1 (46.6)	156.6 (110.1) ^††^	190.2 (110.9) ^###^	<0.001
**White blood cell (×10^9^/L)**	5.4 (1.6)	6.6 (2) ^††^	7.6 (2.2) ^##^	<0.05
**Platelet (×10^9^/L)**	244.1 (59.1)	267.6 (73.2)	242.2 (94.2)	0.28
**Hemoglobin A1c (%)**	8.7 (16.5)	6.4 (1.8) ^††^	7.6 (1.9) ^###,^ **	<0.001
**Glucose (mg/dL)**	86.5 (18.3)	108.3 (31.9) ^††^	129.8 (57) ^###,^ *	<0.001
**Insulin (μU/mL)**	6.8 (3.8)	17.6 (16.1) ^†††^	27.9 (17.6) ^###,^ **	<0.001
**HOMA-IR**	1.6 (1)	4.2 (3.9) ^†††^	7.9 (5.2) ^###,^ **	<0.001
**C3 (mg/dL)**	94.9 (30.5)	125.9 (43.4) ^†††^	154.3 (26.1) ^###,^ *	<0.001
**C4 (mg/dL)**	24.7 (5.5)	28 (8.2)	29.5 (11.4)	0.24
**ELF score**	8.2 (0.8)	8.8 (0.9) ^†^	9.7 (0.8) ^###,^ ***	<0.001
**Liver MRI-PDFF (%)**	3.4 (0.8)	12.6 (6.6) ^†††^	23.2 (10) ^###,^ ***	<0.001
**MRE-LSM (kPa)**	3.1 (0.6)	3.4 (0.7)	5.2 (1) ^###,^ ***	<0.001
**NFS**	−0.4 (0.8)	0.4 (1.6) ^†^	1.4 (2.8) ^##^	<0.05
**FIB-4**	0.7 (0.4)	1.0 (0.7)	2.4 (3.2) ^##^	<0.05

Data are expressed as the mean (SD) or *n* (%), unless otherwise specified. Abbreviations: SBP, systolic blood pressure; DBP, diastolic blood pressure; GGT, gamma-glutamyl transpeptidase; HDL-C, high-density lipoprotein cholesterol; HOMA-IR, homoeostatic model assessment of insulin resistance; C3, complement component 3; C4, complement component 4; ELF, enhanced liver fibrosis. *p* values from the Kruskal-Wallis test are presented in the last column. The statistical significance of each variable of the NASH group compared to the control and NAFL groups was evaluated by post hoc analysis. ^†^, *p* < 0.05; ^††^, *p* < 0.01; and ^†††^, *p* < 0.001 for NAFL vs. healthy controls: ^##^, *p* < 0.01, and ^###^, *p* < 0.001 for NASH vs. healthy control: and *, *p* < 0.05; **, *p* < 0.01; and ***, *p* < 0.001 for NASH vs. NAFL. NA: Not Available.

**Table 2 biomedicines-10-01669-t002:** Plasma levels of six significantly different metabolites in the control, NAFL, and NASH groups.

Plasma Metabolites (ng/μL)	Control (*n* = 25)	NAFL (*n* = 42)	NASH (*n* = 19)	Normalized Values ^a^		Kruskal-Wallis Test	
				NAFL	NASH	*p* Values	Q Values ^b^
Glutamic acid	6.52 ± 2.47	12.74 ± 6.45 ^†††^	15.88 ± 6.36 ^###,^ *	1.95	2.43	<0.001	<0.001
Tyrosine	20.15 ± 9.59	28.21 ± 18.65 ^††^	32.41 ± 19.79 ^###^	1.40	1.61	0.001	0.015
Kynurenic acid	0.005 ± 0.006	0.010 ± 0.009 ^†††^	0.009 ± 0.006 ^##^	1.88	1.70	0.001	0.015
α-Ketoglutaric acid	1.70 ± 0.56	2.63 ± 1.32 ^††^	3.71 ± 1.85 ^###,^ *	1.54	2.18	<0.001	0.004
Myristoleic acid (C14:1)	0.13 ± 0.08	0.22 ± 0.15 ^††^	0.31 ± 0.19 ^###^	1.74	2.50	<0.001	0.004
Palmitoleic acid (C16:1)	3.26 ± 1.87	5.53 ± 3.97 ^††^	7.18 ± 5.16 ^##^	1.69	2.20	0.004	0.048

Data are expressed as the mean ± SD. The full list of plasma metabolites identified in the targeted metabolomics analyses is presented in Appendix A. ^a^, normalized to the corresponding control mean values. ^b^, *p* values adjusted by the FDR. The statistical significance of each variable of the NASH group compared to the control and NAFL groups was evaluated by post hoc analysis. ^††^, *p* < 0.01; and ^†††^, *p* < 0.001 for NAFL vs. healthy controls: ^##^, *p* < 0.01, and ^###^, *p* < 0.001 for NASH vs. healthy control: and *, *p* < 0.05 for NASH vs. NAFL.

**Table 3 biomedicines-10-01669-t003:** The predicting accuracy of the *MetaNASH* score in NAFL and NASH patients with similar BMI, insulin, and glucose levels.

	Group	Mean ± SD	Mean of *MetaNASH* Score	Accuracy	F1 Score
**BMI**	NAFL (*n* = 23)	31.73 ± 2.64	4.326	0.6765	0.6857
	NASH (*n* = 11)	32.26 ± 2.54	5.275		
**Insulin**	NAFL (*n* = 19)	14.52 ± 2.15	4.328	0.7083	0.7742
	NASH (*n* = 5)	13.87 ± 2.21	5.799		
**Glucose**	NAFL (*n* = 33)	95.46 ±8.45	4.305	0.7273	0.7778
	NASH (*n* = 11)	101.0 ± 9.91	5.150		

## Data Availability

The data that support the findings of this study are available on request from the corresponding author. The data are not publicly available due to privacy or ethical restrictions. All parent studies were logged on the website of the national center for medical information and knowledge (NCMIK) (https://cris.nih.go.kr (accessed on 16 June 2020)) in accordance with the International Clinical Trials Registry Platform.

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
