# Peer review of "Plasma Metabolomics and Machine Learning-Driven Novel Diagnostic Signature for Non-Alcoholic Steatohepatitis"

_biomedicines, 2022, doi:10.3390/biomedicines10071669_

Round 1
Reviewer 1 Report
This study proposed new diagnosis parameter for NASH. The work is fine and interesting in that novel MetaNASH score can distinguish NAFL and NASH better than those of conventional scoring including NFS and FIB-4. However, there are still few concerns as follows although they are not serious.
1. The description of line 37-38 and line42-43 seems to be overlapped. Rearrangement would be need.
2. Names of metabolites are showed in legend of Fig.2. However, this is not easy to understand for readers. The lists of the names should be shown on the right side of the figure.
3. In line 391, glycine is not elevated and rather decreased. Additionally, Cis-aconitic acid→myristoleic acid?
4. In Fig5A, what is the number showed as black on white?
5. The process how the formula for calculating MetaNASH was generated should be shown in detail if possible.
Reviewer 2 Report
The article Plasma metabolomics and machine learning-driven novel diagnostic
signature for non-alcoholic steatohepatitis is good one.
Major points
The part of Machine learning and multinomial logistic regression is not clear, please describe it in more details this is the novel part of the paper.
Figure 1 is bioinformatics analysis or machine learning???
Where is the Machine learning algorithims please explainn it in the manuscript in detail
Make the text clear in the figures.
To me machine learning-driven novel diagnostic signature is the novel part of this manuscript. Please work on it and make this is the most important aspect of this paper
Round 2
Reviewer 2 Report
No more comments